# Spatial and Temporal Distribution of Bacterioplankton Molecular Ecological Networks in the Yuan River under Different Human Activity Intensity

**DOI:** 10.3390/microorganisms9071532

**Published:** 2021-07-19

**Authors:** Bobo Wu, Peng Wang, Adam T. Devlin, Lu Chen, Yang Xia, Hua Zhang, Minghua Nie, Mingjun Ding

**Affiliations:** 1School of Geography and Environment, Jiangxi Normal University, Nanchang 330022, China; wbb15279839084@163.com (B.W.); atdevlin@jxnu.edu.cn (A.T.D.); zp1007298060@sina.com (L.C.); xiayang178@163.com (Y.X.); zhangh2013@jxnu.edu.cn (H.Z.); brightchina@163.com (M.N.); dingmingjun1128@163.com (M.D.); 2Key Laboratory of Poyang Lake Wetland and Watershed Research, Ministry of Education, Jiangxi Normal University, Nanchang 330022, China

**Keywords:** human activity intensity, bacterioplankton, molecular ecological networks, freshwater ecosystem functioning

## Abstract

Bacterioplankton communities play a crucial role in freshwater ecosystem functioning, but it is unknown how co-occurrence networks within these communities respond to human activity disturbances. This represents an important knowledge gap because changes in microbial networks could have implications for their functionality and vulnerability to future disturbances. Here, we compare the spatiotemporal and biogeographical patterns of bacterioplankton molecular ecological networks using high-throughput sequencing of Illumina HiSeq and multivariate statistical analyses from a subtropical river during wet and dry seasons. Results demonstrated that the lower reaches (high human activity intensity) network had less of an average degree (10.568/18.363), especially during the dry season, when compared with the upper reaches (low human activity intensity) network (10.685/37.552) during the wet and dry seasons, respectively. The latter formed more complexity networks with more modularity (0.622/0.556) than the lower reaches (high human activity intensity) network (0.505/0.41) during the wet and dry seasons, respectively. Bacterioplankton molecular ecological network under high human activity intensity became significantly less robust, which is mainly caused by altering of the environmental conditions and keystone species. Human activity altered the composition of modules but preserved their ecological roles in the network and environmental factors (dissolved organic carbon, temperature, arsenic, oxidation–reduction potential and Chao1 index) were the best parameters for explaining the variations in bacterioplankton molecular ecological network structure and modules. *Proteobacteria*, *Actinobacteria* and *Bacteroidetes* were the keystone phylum in shaping the structure and niche differentiations in the network. In addition, the lower reaches (high human activity intensity) reduce the bacterioplankton diversity and ecological niche differentiation, which deterministic processes become more important with increased farmland and constructed land area (especially farmland) with only 35% and 40% of the community variation explained by the neutral community model during the wet season and dry season, respectively. Keystone species in high human activity intensity stress habitats yield intense functional potentials and Bacterioplankton communities harbor keystone taxa in different human activity intensity stress habitats, which may exert their influence on microbiome network composition regardless of abundance. Therefore, human activity plays a crucial role in shaping the structure and function of bacterioplankton molecular ecological networks in subtropical rivers and understanding the mechanisms of this process can provide important information about human–water interaction processes, sustainable uses of freshwater as well as watershed management and conservation.

## 1. Introduction

River ecosystems provide important support for terrestrial and aquatic ecosystems and provide important services for human health, well-being as well as economic and social benefits [1]. The bacterioplankton community is especially sensitive to environmental change, which is a ubiquitous and indispensable freshwater river ecosystem component that plays a key role in biogeochemical processes [2]. The structure of the bacterioplankton community can also reflect the ecological environment of the river to a certain extent and is an ideal indicator that can be used to monitor the ecological impacts of human activities on the functional characteristics of the river water environment [3]. Previous work has shown that human influences affect microbiome composition [4,5,6], microbe–microbe interactions [7,8] and microbe–host interactions [3,9,10]. Although it is well documented that such changes in network structure affect ecosystem functioning and stability, little is known about the link between human activity intensity and the stability of these microbial systems and whether and how the ecological networks, particularly bacterioplankton molecular ecological networks, will change under human activity intensity change scenarios.

To meet human needs, human activities directly affect land use and change landscape pattern [11,12]. As a hydrological connection between terrestrial systems and coastal systems, rivers are intimately associated with surrounding changes in land use and play an important role in the biogeochemical cycle [13]. The conversion of natural vegetation to anthropogenic land uses (e.g., urban expansion and agriculture) is often accompanied by increases in impervious surfaces as well as the fragmentation of natural vegetation [14]. An increased level of nutrient loads and no-point pollutants from domestic sewage, industries, and agriculture can considerably affect the water environment conditions [15], disturb the spatial distributions of bacterioplankton population, communities and habitats [16], reduce microbial diversity [17], and destabilize microbial co-occurrence networks interactions (e.g., through predation and growth competition) [18]. These findings highlight the fact that rivers are among the most vulnerable ecosystems in the context of a growing human population, much of which is often concentrated along the riverside, and increasing anthropogenic pressure. Furthermore, the ecological mechanisms controlling microbial community assembly and interspecies interactions in anthropogenically disturbed rivers have not been resolved with respect to complex abiotic and biotic environmental factors. Understanding the response of river bacterioplankton molecular ecological network and functioning to human activity is critical to human well-being and river sustainable management.

Network analyses have been used to explore the ecological interaction patterns among microbial species in oceans [19], rivers [20], lakes [21] and soils [22], which can reveal complex associations within microbial communities [23]. Properties of ecological networks, which might represent interactions between co-existing organisms, can influence the response of communities to environmental change, including human activity [24,25,26,27]. The topological properties obtained from network analysis can be used to define network complexity or stability between microbial community and environmental factors [28,29]. Network analysis also can help identify potential keystone species [30]. These keystone taxa can help disentangle microbial co-abundance and provides comprehensive insights into the microbial community structure and assembly patterns [31,32]. Additionally, each network can be partitioned into ecological clusters of exclusive taxa, here denoted “modules”, which allow more robust statistical inferences by integrating higher-dimensional data into predictable ecological clusters [33]. Despite an increasing use of network analysis in ecology [6,24,34], our understanding of co-occurrences or potential interactions within complex bacterioplankton, studies focused on microbial keystone species among multiple bacterioplankton in subtropical freshwater ecosystems and how bacterioplankton networks respond to disturbances such as human perturbation (such as land-use change), remains scant.

The Yuan River is a tributary of the Ganjiang River, which is the largest tributary of Poyang Lake, China. The Yuan River has diverse riparian habitats from upstream to downstream, including forest, mountains, cities, and even a reservoir along streams in the middle reaches. Previous studies have determined that the Yuan River contains abundant nitrogen and phosphorus pollution [35]. Moreover, chromium (Cr), iron (Fe) and arsenic (As) in the river mainly come from urban sewage industrial activities and mining activities [36]. Xu et al. [37] evaluated the influence of landscape structures, including structural composition and spatial configuration, on river water quality at scales ranging from riparian zones to entire watersheds. Previous efforts (e.g., Zhao et al. [16]) have investigated the role of environmental factors in shaping bacterioplankton communities in this study area. However, the effect of human activity intensity on the spatial and temporal distribution of bacterioplankton molecular ecological networks is still unclear. Consequently, we hypothesized that human activity intensity affect bacterioplankton co-occurrence patterns, as reflected by changes in topological properties, keystone species, and module composition, and these changes would increase with the intensification of human activities. Our objectives were to (i) investigate the effect of human activity intensity on the topological properties of bacterioplankton molecular ecological networks; (ii) identify keystone species and evaluate their relationships with water chemistry parameters; and (iii) evaluate how human activity intensity affects the composition of each module within the bacterioplankton molecular ecological network.

## 2. Materials and Methods

### 2.1. Study Area, Sampling and Physiochemical Analysis

The Yuan River basin (N 27°33′~28°05′, E 113°54′~114°37′) is located in the Jiangxi Province, southeastern China, originating in the western foothills of the Wugong Mountains and ultimately discharging into the Ganjiang River (Figure 1). The watershed covers a total area of 6262 km^2^ and has a total length of 279 km (the mainstream is indicated by “Y” in Figure 1). Upstream reaches of the river (Y01–Y05) have substantial forest cover. The middle reach of the river (Y06–Y11) has a large-scale water project (Jiangkou Reservoir), with a storage capacity of 320 million m^3^. Downstream (Y12–Y16) is mainly farmland, and here is a crucial industrial base of Jiangxi province. The Yuan River basin is situated in a humid subtropical monsoon climate zone. The average annual precipitation in the basin is 1583 mm, with the rainfall in April to June accounting for 45% of the total. 

In this study, we gathered surface water samples from 16 sites in the mainstream of the Yuan River. The surface waters of the river at a depth of 50 cm were collected in August 2018 and January 2019, representing the wet and dry seasons, respectively. The samples were filtered through a 0.45 μm acetate filter membrane, placed into a sealed sampling bottle, and then refrigerated at 0–4 °C. Field determinations of dissolved oxygen (DO), oxidation–reduction potential (ORP), formazine nephelometric unit (FNU), electric conductivity (EC) and pH were performed using a portable water quality analyzer (HI9828, Hanna Instruments Ltd., Rome, Italy); an automatic discontinuous analyzer (Smartchem 200 Brookfield, WI, USA) for determination of ammonia nitrogen (NH4^+^-N), nitrate–nitrogen (NO_3_^−^–N) and total phosphorus (TP); and chlorine and sulfate ions (Cl^−^, SO_4_^2−^) were determined using an Ics-2100 ion chromatography system. Dissolved organic carbon (DOC) was measured with a TOC analyzer (Shimadzu TOC-L CPH, Kyoto, Japan). Trace metals, including aluminum (Al); Cr; vanadium (V); molybdenum (Mo); titanium (Ti); manganese (Mn); uranium (U); Fe, cobalt (Co); nickel (Ni); cuprum (Cu); zinc (Zn); As; cadmium (Cd), and lead (Pb), were measured with ICP–MS (Thermo X series II, NE, USA). Kalium (K), calcium (Ca), sodium (Na) and magnesium (Mg) were measured with ICP-AES (Optima 8000, PerkinElmer, Waltham, MS, USA).

Digital elevation model (DEM) data (at a 30 m resolution) were used to delineate basin boundaries. Sub-basin classifications at each sampling site ranged from a single sampling site that encompassed the sub-basin area to the inclusion of adjacent upper sample sites to reflect the inputs of allochthonous bacteria due to the fast population growth and replacement rates of bacterial communities.

Landsat 8 satellite imagery from 2017 was used to generate a land-cover classification at 30 m resolution. Land use pattern images were obtained from the Geospatial Data Cloud. Images were then binned into five classes: farmlands, forests, freshwaters, urban areas and others. ArcGIS v 10.3 was used to delineate basin boundaries and calculate land use proportions.

### 2.2. DNA Extraction and Illumina DNA Sequencing

Samples were pre-filtered through a 5 μm Durapore membrane filter (diameter 25 mm; Xinya, China) to remove particulates and algal biomass, followed by filtering through a 0.22 μm Durapore membrane filter (diameter 25 mm; Xinya, China) to collect microbial cells. Each water sample was simultaneously filtered through several filters to reduce filtering time. Filters from each sample were mixed and stored at −80 °C for subsequent DNA extraction.

Total DNA was extracted from the water samples using the E.Z.N.A.^®^ Soil DNA Kit (Omega Bio-tek, Norcross, GA, USA). The bacterial V4–V5 hypervariable regions of 16S rRNA genes were amplified using the forward primer 338F (5′-ACTCCTACGGGAGGCAGCA-3′) and reverse primer 806R (5′-GGACTACHVGGGTWTCTAAT-3′) [16]. PCRs were conducted using the following PCR cycling parameters: initial denaturation for 2 min at 95 °C, followed by 25 cycles of 30 s at 95 °C, annealing for 30 s at 55 °C, and elongation for 30 s at 72 °C, all followed by a final elongation step for 5 min at 72 °C. Gel electrophoresis on 2% agarose gels was used to ensure to evaluate successful PCR amplification. Triplicate PCR amplicon products were pooled for each sample, purified using an AxyPrep DNA gel extraction kit (Axygen, Corning, NY, USA), and quantified using the QuantiFluor™-ST system (Promega, Madison, Wi, USA). DNA sequencing was conducted on the Illumina MiSeq platform (Illumina, San Diego, CA, USA) following standard operating procedures and paired-end 2 × 250 bp sequencing chemistry. The sequencing was conducted at the Shanghai Majorbio Bio-Pharm Technology Co., Ltd. of China. Raw sequence data files were deposited in the NCBI Sequence Read Archive database (Accession number: SRP194014).

### 2.3. Statistical Analyses

Operational taxonomic units (OTUs) were clustered with a 97% similarity cutoff using UPARSE (version 7.1) and chimeric sequences were identified and removed using UCHIME. The phylogenetic affiliation of each 16S rRNA gene sequence was analyzed by the RDP Classifier (Release11.3) against the Silva (Release 119) 16S rRNA database using a confidence threshold of 70%. Dilution curve analysis was performed based on OTU. We evaluated the alpha-diversity through the Chao1 richness index and the Shannon diversity index. All the data were tested for normality (Shapiro–Wilk test). Variables that were not normally distributed were log transformed to normality. Pearson correlations and one-way analyses of variance (ANOVA) with Fisher’s least significant difference (LSD) post hoc tests were performed using SPSS Statistics v20. The LSD method was used for multiple comparison, and Pearson correlation analysis was used for correlation analysis (with a significance level of *p* ≤ 0.05 considered as a significant difference). A Venn diagram was constructed to reflect the number of common and unique OTUs among two bacterial molecular ecology networks.

To estimate the potential contribution of stochastic processes to bacterioplankton communities, a neutral community model was used [38]. The parameter R^2^ was used to indicate the overall fit to the neutral model, while *m* represents the immigration rate. R^2^ > 0 means that the population conforms to the neutral model (stochastic processes), while R^2^ < 0 indicates the opposite. Please refer to Mo et al. [39] for the calculation process of the neutral model.

Bacterioplankton molecular ecological networks were constructed by the 16Sr DNA and molecular ecological network technologies. Network analysis (based on phyla level) was performed to identify the interrelations between microbial taxa, using Cytoscape version 3.4.0 combined with the CONET plug-in (http://apps.cytoscape.org/apps/conet, accessed on 1 July 2021) [21]. Cytoscape (version 3.4.0) was used for network visualization and modularization analysis to determine associations (positive and negative correlations) between bacterial community members [22]. To highlight the most important interactions, only strong positive (*r* > 0.8) and strong negative (*r* < −0.8) relationships were shown in the network diagrams [17]. The Network Analyzer tool was used to calculate the network topology parameters, such as average clustering coefficient (avgCC), average path distance (APD), average degree (avgK) and centralization of degree (CD). The classification identifier for each OTU was assigned at the category level. The resulting original OTU table (one with taxonomic abundance) was used as the input matrix. The network was built according to the guidelines provided on the CONET website (http://psbweb05.psb.ugent.be/conet/tutorial4.php, accessed on 1 July 2021). Parameters were set as follows: at least 30 sequences are preprocessed and filtered for each OTU, and there are four similarity measures (Spearman, Pearson, Kullbackleibler and Bray–Curtis) and an automatic threshold setting. The error detection rate (FDR) correction was set to 0.05 (*p* < 0.05).

In addition, the *z_i_*-score and *p_i_*-score cut-offs were based on the methods of metabolic networks [40]. Here, we define nodes as network hubs (*z_i_*-score > 2.5; *p_i_*-score > 0.62), module hubs (*z_i_*-score > 2.5; *p_i_*-score < 0.62), connectors (*z_i_*-score < 2.5; *p_i_*-score > 0.62) and peripherals (*z_i_*-score < 2.5; *p_i_*-score < 0.62), based on their within-module degree (*z_i_* -score) and participation coefficient (*p_i_*-score) threshold value [41], which determines how each node is positioned within a specific module or how it interacts with other modules [9]. The network hubs were highly connected, both in general and within a module, the module hubs were highly connected within a module, the connectors provided links among multiple modules and the peripherals had few links to other species [41]. Network hubs, module hubs, and connectors were termed keystone network topological features; these are considered to play important roles in the stability and resistance of microbial communities [42]; thus, we define the OTUs associated with these nodes as keystone species.

Based on the perspective of the land use/cover concept, the human activity intensity of land surface is defined as the degree of natural cover use, transformation, and exploitation of land surface by humans in a certain region [11]. This degree can be reflected by land use/cover types. Obviously, the human activity intensity of land surface belongs to the conceptual category of the general human activity intensity, which refers to the influence of economic and social activities on a regional natural complex. The use, transformation, and exploitation of land surface can be seen as the main body of human activity, but not the whole. Therefore, this paper determines that the human activity intensity is the overall degree of human interference to the surface. Direct human interference types mainly include farmland and residential land. Human activity intensity of land surface can be expressed as
$$\mathrm{HAILS}=\frac{Si}{S}\times 100\%$$
where HAILS is human activity intensity of land surface; *S**i* is the area of human activity-influenced land; *i* is the cover type; and *S* is the total land area.

## 3. Results

### 3.1. Quantification of the Human Activity Intensity

We examined the spatial distribution of farmland, forests, water, residential land and other areas, to quantify the effects of human disturbance. Forest and farmland are the dominant land cover types in the watershed. Forest areas significantly decreased (*p* = 0.001) in the lower reaches of the river while farmlands increased significantly (*p* = 0.002). In order to further analyze the human activity intensity of land surface, the change in the HAILS parameter was calculated, as shown in Appendix A. The results show that the Yuan River watershed under different human activity intensity has a gradient from upstream to downstream and was divided into three parts according to the HAILS level: (1) upper reaches (low HAILS level); (2) middle reaches (middle HAILS level); and (3) lower reaches (high HAILS level).

### 3.2. Taxonomic Diversity of the Bacterioplankton Community in the Yuan River

After quality filtering and subsampling (57,068 reads per sample), a total of 3945 bacterial OTUs included here belonged to 46 taxonomic groups in the soil samples. All OTUs were assigned to 1619 species, 847 genera, 392 families, 213 orders, 118 classes and 46 phyla. There are differences in the number of OTU species between wet and dry seasons and the number of species obtained at various taxonomic levels (Appendix A). Good’s coverage for the observed OTUs was 99.21 ± 0.61%, indicating a near-complete sampling of the community.

The Chao1 richness index was significantly higher in the upper reaches than in the lower reaches (*p* = 0.022) in the dry season communities (Appendix A). During both the wet season and the dry season, there were no significant differences in the Shannon diversity index between the upper, middle and lower reaches.

PLS-DA analysis clearly distinguished three groups of wet season communities and three groups of dry season bacterioplankton communities that were based on the different reaches, with the first two axes explaining 29.98% and 9.99% of the total variation in during the wet season and 17.02% and 12.59% during the dry season, respectively (Appendix A). Due to a limited sampling number, there is a limited explanation to the observed variability. Additionally, we observed significant differences (especially the upper and middle reaches) in bacterioplankton communities between different sample sites during both the wet season and the dry season.

The neutral community model (NCM) successfully estimated a large fraction of the relationship between the occurrence frequency of OTUs and their relative abundance variations (Appendix A). These results indicated that the explained variation of NCM tended to remain relatively large and consistent and the relative importance of stochastic processes decreased in the following order: upper reaches (R^2^ = 0.615/0.563, m = 0.6882/0.8151) > middle reaches (R^2^ = 0.439/0.462, m = 0.2266/0.541) > lower reaches (R^2^ = 0.35/0.4, m = 0.4525/0.6514) during the wet and dry seasons, respectively.

### 3.3. Taxonomic Composition and Functional Analysis of the Bacterioplankton Community in the Yuan River

The most common phyla identified in the water samples were *Actinobacteria, Proteobacteria* and *Bacteroidetes* (Figure 2). The majority of bacterial sequences identified during the wet season belonged to *Actinobacteria* (35.14%); these bacteria were significantly more abundant during the wet season than during the dry season (28.89%). During both the wet season and dry season, *Actinobacteria* abundance was significantly lower in the upper reaches, as compared to either the middle reaches or the lower reaches. *Proteobacteria* was the most abundant phylum during the dry season (33.41%) and the second largest phylum during the wet season (33.88%), which showed no significant difference between the dry season and the wet season. The abundance of *Proteobacteria* was significantly higher in the upper reaches (as compared to the lower and middle reaches) during the wet season. *Bacteroidetes* were the third most abundant phylum during the dry season (28.23%); these bacteria were significantly more abundant during the dry season than during the wet season (17.56%). During the wet season, *Bacteroidetes* were significantly more abundant in the upper reaches than in the middle and lower reaches. In contrast, *Proteobacteria* and *Bacteroidetes* abundance was stable across all of the sites during the dry season. In addition, *Proteobacteria* and *Cyanobacteria* abundance fluctuated among sampling sites in during the wet season. *Cyanobacteria* and *Verrucomicrobia* were significantly more abundant during the wet season (6.22% and 3.77%, respectively) than during the dry season (1.56% and 1.33%,respectively). During the wet season, *Cyanobacteria* were significantly more abundant in the middle reaches than in the upper and lower reaches. In the lower reaches, *Verrucomicrobia* abundance was significantly lower during both the dry season and the wet season, as compared to either the upper reaches or the middle reaches.

The relative abundance of the PICRUSt inferred function is illustrated in Appendix A. Compared to taxonomic composition, the functional analysis of all samples was similar during both the wet season and dry season. Amino acid transport and metabolism, cell cycle control, cell division, chromosome partitioning, general function prediction only and function unknown were the most abundant functions in all samples.

### 3.4. Interactions between Bacterial Taxa in the Network

The six networks of bacterial communities revealed distinct co-occurrence patterns (Table 1 and Table 2). These nodes belong to 31 bacteria phyla, where *Proteobacteria,*
*Actinobacteria Bacteroidetes* and *Parcubacteria* were primarily found (Figure 3). Furthermore, *Proteobacteria* was the dominant phyla in bacterial networks and was not altered by spatial and temporal distribution.

All the bacterial networks during the dry season were larger, more connected, more modular and had more negative correlations than during the wet season. Compared to the upper and middle reaches, the ratio of negative/positive and modularity in bacterial networks decreased in the lower reaches, indicating that the bacterial network here was more robust during the wet season. However, there were no significant differences between the upper and lower reaches networks during the wet season as measured by the number of nodes (TNs), links (TLs), and avgK. In contrast, Figure 3 shows that the network in the upper reaches were much more complex and harbored more dominant phyla than other networks during the dry season. TNs in the upper reaches network increased during the dry season compared to the middle and lower reaches; the lower reaches network was 48.1% lower than the upper reaches network. Although the ratio of negative/positive (NP) in the upper reaches network was the lowest, TLs were the largest during the dry season. In particular, *Proteobacteria**, Actinobacteria Bacteroidetes* and *Parcubacteria* were more intricately linked, indicating that the upper reaches promote the interaction (especially competition) between bacterial taxa. In addition, as compared to the upper and middle reaches, the lower modularity (M) and avgCC in the lower reaches resulted in decreasing interactions of bacterial taxa during the dry season.

The overlapped and unique OTUs in the bacterioplankton molecular network are illustrated by a Venn diagram (Figure 4). The upper, middle and lower reaches shared 123, 72 and 99 OTUs by the two seasons, which accounted for 10.7%, 10% and 13.3% of the total bacterial OTUs, respectively. The results indicated that different seasons and reaches significantly affected the distribution of species in the bacterial ecological networks of the Yuan River.

We also calculated how co-occurrence network complexity was correlated with environmental variables (Figure 5). We observed that DOC, T, and As were significantly negatively correlated with avgK, negative links (NLs), positive links (PLs), TLs and TNs, while correlations with ORP, Mn and the Chao1 index were positive. Similarly, SO_4_^2−^ was significantly negatively correlated with NLs, TLs, and TNs. However, the co-occurrence network was uncorrelated with any nutrient variables factors except for TP, which was significantly negatively correlated with graph density (GD).

### 3.5. Keystone Species in Bacterial Networks

We defined the hubs and connectors as keystone species, by which we mean that if these taxa were removed, the modules and networks may also disassemble. During both the wet season and the dry season, no node in the bacterial network in the upper, middle, and lower reaches falls in the network hubs and more than 78% of the nodes fall in the peripheral modules; the remaining nodes are classified as module hubs and connectors (Figure 6). The proportion of module hubs (wet season: 0%, 0%, 1.26%; dry season: 0.3%, 0.19%, 1.53%) and connectors (wet season: 7.69%, 7%, 14.83%; dry season: 12.97%, 11.42%, 19.66%) increased from the upper reaches to the lower reaches while the proportion of peripheral modules (wet season: 92.31%, 93%, 83.91%; dry season: 86.73%, 88.39%, 78.82%) decreased, indicating a more hub-based and connected network structure in the lower reaches.

The node with the highest connectivity among the keystone species in the module hubs and connectors is completely different between the wet season and the dry season. Some of these had low relative abundance. In the upper reaches, there was one single connector during the wet season belonging to *Gemmatimonadetes* and two connector OTUs during the dry season belonging to *Proteobacteria* and *Ignavibacteriae*. In the middle reaches, there was one single connector OTU belonging to *Proteobacteria* during the wet season and one single connector OTU during the dry season belonging to *Bacteroidetes*. For the lower reaches, there were one single connector during the wet season belonging to *Verrucomicrobia* and two module hub OTUs during the dry season belonging to *Bacteroidetes* and *Firmicutes*.

The number of edges that linked network hubs, module hubs and connector nodes with functional nodes reflects the linkage between keystone species and functions. Generally, the middle reaches only presented overall lower average degrees during the dry season (Appendix A). Conversely, stronger connections indicated by the number of edges between the nodes were observed in other networks, especially in the lower reaches and the upper reaches during the dry season. However, fewer of the module hubs presented weak connections with functions, while connectors had strong connections with functions and occupied nearly all of the connections.

We constructed new networks by correlating the relative abundance of keystone species with the water chemistry parameters (Appendix A). When significant correlations were present, the strength of correlations increased from the upper reaches to the lower reaches, especially during the dry season. Furthermore, DOC, EC, HCL^−^, ORP, Ca, NO_3_^−^-N, Cr and Mn were correlated with more keystone species than other water properties. When considering the correlated links, the proportion of negative correlations between keystone species and water properties was greater than the number of positive correlations, and the negative correlations were numerically greater in the lower reaches as compared to the upper and middle reaches.

### 3.6. Major Modules in Bacterial Networks

Modules are groups of nodes that are well connected with one another but less connected with nodes belonging to other modules [43]. To identify assemblages that potentially interact or share niches within bacterioplankton, we focused on major modules (node number > 5% of the total nodes) (Figure 7). There were direct interactions between modules except in the middle reaches, and most of the edges of networks were intra-modular. Fewer direct interactions were detected in the wet season networks, as taxa tended to co-occur (positive correlations, green lines) rather than co-exclude (negative correlations, red lines). The proportion of inter-module edges was higher in upper reaches, indicating that modules were more isolated in the middle and lower reaches during both the wet season and dry season.

The composition of modules differed within each network and changed over time. *Proteobacteria**, Actinobacteria* and *Bacteroidetes* (especially *Proteobacteria*) were mostly distributed in the modules of the wet season and *Proteobacteria**, Parcubacteria, Firmicutes* and *Bacteroidetes* (especially *Proteobacteria*) were mostly distributed in the modules of the dry season. These results indicate that specific modules might occupy distinct niches or perform unique functions, while human activity did not change their ecological roles. Interestingly, DOC, T, As and Cr were significantly negatively correlated with TL, while ORP, DO and the Chao1 index was positively correlated. Similarly, SO_4_^2−^ was significantly negatively correlated with TL (Appendix A).

## 4. Discussion

### 4.1. Human Activity Mediates the Assembly Processes of Bacterioplankton Communities via Altering Environmental Conditions

Human activity intensity has an important influence on the assembly of all bacterioplankton communities, primarily by affecting the balance between deterministic and stochastic processes. The degree to which deterministic vs. stochastic processes shape the bacterioplankton community in the Yuan River is determined more by human activity intensity rather than by seasonality. Our results suggested that stochastic and deterministic processes were simultaneously responsible for shaping the bacterioplankton communities in the Yuan River. These results are consistent with the findings of numerous previous studies, where the stochastic and deterministic processes were jointly credited for the assembly of bacterioplankton communities [44,45]. However, the relative importance of deterministic and stochastic processes varied in different hydrological seasons and reaches. The community variation explained by stochastic processes decreased from 61.5%/56.3% at the high-human activity intensity level to 35%/40% at low-human activity intensity level during the wet and dry seasons, respectively (Appendix A). In other words, the relative importance of deterministic processes (e.g., environmental filtering) tended to increase from the upper reaches to the lower reaches and the lower reaches bacterioplankton communities were more likely to be affected by the changes in the riverine environmental conditions. Increased human activities along rivers may not only increase the quantities of nutrients, but also change the forms and proportion of nutrients and physicochemical variables [46]. In addition, the HAILS level (Appendix A) and the strength of correlations between the relative abundance of keystone species with water chemistry parameter (Appendix A) increased from the upper reaches to the lower reaches, implying that the community assembly was more strongly influenced by deterministic processes at the high-human activity intensity level.

### 4.2. Human Activity Destabilizes Bacterioplankton Molecular Ecological Network Stability in the Yuan River

Human activity decreased the bacterioplankton interactive stability in the network. Negative links might stabilize co-oscillation in communities and promote the stability of networks [28]. Stability is promoted by limiting positive feedbacks and weakening ecological interactions. There were higher negative/positive ratios seen the upper reaches during the wet season, which indicates that hosts can benefit from microbial competition when this competition dampens cooperative networks and increases stability. However, when the lower reaches with the lowest negative/positive ratio are perturbed by the external environment, the interaction network of the bacterioplankton community will transmit the environmental perturbation to the entire network in a short time and result in an unstable network structure. At the same time, this unstable network may lead to significant changes in the bacterial community involved in amino acid transport and metabolism, cell cycle control, cell division, chromosome partitioning and other functions [47], which in turn affect river ecosystem functions and stability. Meanwhile, the upper reaches bacterioplankton molecular ecological networks formed a much more complex (i.e., higher connectivity) and larger (i.e., more nodes and links) network during the dry season. In this way, the increased complexity of the network structure in the upper reaches may lead to higher community stability with a mixed interaction, increase the breadth of the niche, strengthen the interconnections between different bacteria in the bacterioplankton food web, enhance the efficiency of resource transfer and help it use water nutrients more effectively; this result is similar to that reported by Zhang [48]. Therefore, although the ratio of negative/positive in the upper reaches network was the lowest, the microbe (especially *Proteobacteria**, Actinobacteria Bacteroidetes* and *Parcubacteria*) can respond to environmental pressures by reducing competition and strengthening cooperation during the dry season. Additionally, a higher modularity and clustering coefficient indicated a marginally higher modularity [49]. An increase in the number of modules indicates more niches and network modularity also can enhance the stability of the network under human activity disturbances [50]. Our network analyses showed that compared to the upper and middle reaches, the lower modularity and clustering coefficient yielded less bacterial links in the lower reaches, an outcome perhaps related to the frequency of competitive interactions decreases and that of facilitative interactions increases as stress intensifies. Meanwhile, modularity measures the connectivity between nodes within their own modules that would not occur by chance [51]. Although the connectors and module hubs were more likely to cope with environmental perturbation through mutual cooperation (Appendix A), the interaction network of the lower reaches was easily perturbed by the external environment, and the resource competition within the bacterial community was weak and the stability of the interaction of the bacterial community was poor.

The copiotrophic *Proteobacteria* dominated during the dry season, providing evidence for the existence of distinct and discrete ecological niches over temporal scales (i.e., modules) in the river ecosystem that are preferentially occupied by different groups of microbial taxa. We found that the predominant taxa (especially *Proteobacteria*) drive network structure in the upper reaches or the middle reaches. Compared with other predominant bacterial phyla, *Proteobacteria* with copiotrophic advantages may demonstrate wider niche breadths and higher anti-interference capacities and play a dominant role in maintaining the stability of the bacterioplankton community interaction networks. Other less dominant phyla can be used as a diversified library to enhance the resilience of microbial communities and resistance capacities to environmental perturbations. While the relationships between bacterial phylogeny and function are complex, shifts in the abundance of indicator taxa might inform us of the stability of these networks, and consequently, on the response of bacterioplankton communities to human activity. In the upper reaches, we also found that *Actinobacteria*, which are well known as “ultramicrobacteria” that prefer oligotrophic niches [52], were significantly lower during both the dry season and the wet season, as compared to the middle reaches and the lower reaches. The “competitive” taxa that engage in many antagonistic interspecific interactions are replaced by slow-growing, stress-tolerant species (e.g., oligotrophic microbes) as stress increases [6], indicating that the bacterial network under the upper reaches was more robust during both the wet season and dry season.

Abiotic and biotic factors showed significant correlation with network indexes in bacterioplankton co-occurrence patterns. DOC, T, As, ORP, Mn, SO4^2−^ and TP were significantly correlated with network properties, while the increased network size and connectivity was accompanied by increasing bacterial Chao1 diversity (Figure 5). The effect of pollution on the bacterioplankton community structure can be translated into environmental filtering. Therefore, the less complex network in the lower reaches during the dry season may be caused by the combined human activities of sewage discharge, urban runoff and other anthropogenic contaminants. Decline in bacterioplankton diversity was closely related to human activity such as increasing agricultural land use, impervious surface cover, housing density and urban population pressure. These results indicate that although the lower reaches during the wet season increased the number of positive and negative correlation links in the bacterial network, the Chao1 index was the lowest. These stronger negative interactions between only a few species under the lower reaches exclude more species from the community and result in a loss of biodiversity (Appendix A). At the same time, these stronger interactions also decrease the stability of the bacterial communities, providing a mechanistic link between species interaction, biodiversity and stability.

### 4.3. Human Activity Increase Triggers Keystone Species Change

Different keystone species may play the same functional role in different human activity intensity and less abundant taxa can be as important as abundant ones in maintaining microbial networks. Module hubs and connectors are often treated as keystone species, as they have disproportionately important roles in maintaining a network structure relative to the other taxa in the network. The disappearance of these keystone taxa may cause modules and networks to disassemble and thus keystone taxa may play a role in maintaining ecosystem stability [53]. In this study, the keystone species in the three reaches are mostly different, with the middle and lower reaches during the wet season being the only exception (Figure 6). Seasonal variability determines the structural and compositional properties of microbiomes in an environment, and as such, a keystone species might only be present in a specific season or time period [54]. Similarly, the keystone species were assigned to diverse OTUs, indicating that the topological roles of individual OTUs and keystone species can be altered by human activity. A few taxa acted as hubs or connectors in two different networks, which suggests that the conditions present were not identical in terms of human activity intensity and supports the context dependency theory that keystone species play critical roles only under certain conditions [55]. Lupatini et al. [56] also found that the keystone species changed as conditions changed. However, changes in keystone species are not necessarily associated with changes in bacterioplankton function. The general functional structures of all samples were similar to one another in the present study (Appendix A). Alternatively, the unique keystone taxa detected in the river networks could be explained by functional redundancy [57]. Given the possibility that most microbes inhabiting in river possessed similar functional genes, the fluctuation in taxonomic structure along human activity intensity stress habitats would not necessarily alter the microbial function structure. Such an ability could serve as a fundamental property of bacterioplankton which is essential to environmental perturbation. The similar weak linkages between microbial taxonomic and functional community structure were previously observed in microbial stream biofilms [58], which further corroborates the theory. In addition, most of the bacterial keystone species belonged to *Proteobacteria* and *Bacteroidetes*, which is consistent with other studies [45]. The dominate phyla often affect ecosystem exclusively by virtue of sheer abundance [54]. Some of the highest connectivity among the keystone species had relatively low abundance, which suggests that low abundance taxa (*Gemmatimonadetes, Ignavibacteriae, Firmicutes* and *Verrucomicrobia*) may play important roles in maintaining network structures in bacterioplankton communities. Zeng et al. [59] found that *Gemmatimonadetes* contained chlorophyll-based phototrophic species, suggesting a strong ability to support fundamental biological processes. *Ignavibacteriae* were reported to be core populations that endowed the bacterial community with stronger dechlorination and phenol-degradation abilities [60]. Not only are *Firmicutes* able to withstand resource stress via the formation of endospores, but they are also able to adapt to resource-rich conditions [61]. A recent freshwater metagenomic study of 19 *Verrucomicrobia* suggest that members of this phylum act as polysaccharide degraders in freshwater systems [62]. These studies support the idea that although the species are rare, they are likely to provide complementary or unique metabolic pathways to service the ecosystem.

In addition, keystone species in higher human activity intensity stress habitats provide intense functional potentials. Generally, stronger connections indicated by the number of edges between nodes were observed in keystone species and functional networks, especially in the lower reaches (Appendix A). On the contrary, less of the connectors or module hubs presented a strong connection with functions in other networks (Appendix A). This means they may play fundamental roles in monitoring the functioning of multiple ecosystem processes in higher human activity intensity stress habitats. However, the influence of water chemistry on bacterioplankton community networks may be due to the spatial distribution of land use patterns. There were more significant correlations between keystone species and water chemistry parameters in higher human activity intensity stress habitats (Appendix A). The network is more influenced by keystone species with greater sensitivity to water chemistry parameters after disturbance, and once disrupted, the ecosystem will have difficulty recovering.

### 4.4. Human Activity Reduces Ecological Niche Differentiation and These Interacts Less with Each Other in the Yuan River

The potential for extensive mutualistic interactions exists among bacterioplankton in river assemblages, especially in the upper and middle reaches. We identified modules within the networks that likely result from bacterioplankton–bacterioplankton interactions or covariation in response to shared niches in the river. Modules have been treated as niches, and each of the modules identified in the network reflected species’ environmental preferences and habitat heterogeneity [55]. In our study, most interactions were intra-module and the edges occurring within modules were predominantly positive. This suggests that the microbial taxa within the same module might form cooperative interactions or share similar guilds or niches. In addition, our results indicated that there was lower modularity in the lower reaches (Table 1 and Table 2). Nodes in the same module occupy similar ecological functions and niches [63]. Hence, the bacterioplankton community could hold more diverse but scattered niches in the upper and middle reaches. Meanwhile, MCODE analysis showed that the major modules in the upper and middle reaches formed considerably larger complex networks as indicated by the values of TNs and TLs (Appendix A), indicative of the centralized functions and niches in the community operating to maintain water ecosystem stability. The functional specialty of modules was to minimize the impact of environmental turbulence [64]. Thus, more inter-module edges were detected in the upper reaches network (especially during the dry season), which would ensure quicker communication and more efficient regulation between its modules in response to environmental stimuli. Alternatively, fewer inter-module edges in the lower reaches suggest that modules in this network were more isolated. Convergence in microbial functions due to the influence of human disturbance is likely to disturb microorganism dispersal and occur over larger spatial scales owing to the degree of anthropogenic disturbance in streams, resulting in less ecological interactions or niche sharing [65].

Human activity altered the composition of modules, but preserved their ecological roles in the network, and the critical importance of *Proteobacteria,*
*Actinobacteria* and *Bacteroidetes* in shaping the structure and niche differentiations in the bacterioplankton molecular ecological networks was revealed. This may also have been explained by the insurance hypothesis and functional redundancy theory [43]. Considering that highly connected species from the same module might share similar ecological characteristics or play similar functional roles in ecosystem stability, it is possible that they are ecologically redundant. Members of a module ultimately declined as a result of water chemistry parameter alteration after disturbance. However, these declining organisms can be replaced by biologically unique but ecologically equivalent species, which then refill their unoccupied niches [66]. This cycling of species might result in changes in module composition but would preserve their ecological roles that are necessary to sustain the ecosystem [67].

Since the abundance and community distribution of microorganisms alone cannot characterize the internal relationship between them [68], this paper analyzed the correlation of water chemistry parameters from the perspective of molecular ecological network. We found that the ORP and the Chao1 index were both significantly positively correlated with the molecular ecological network structure and modules of the Yuan River, while DOC, T and As were negatively correlated. Previous studies have observed that DOC is a significant driver of microbial community composition in rivers [69,70,71]. DOC can come from extracellular release and leachate from phytoplankton and macrophytes and soil flow pathways and may be enriched by domestic sewage or agricultural runoff [16]. There is evidence that bacterioplankton communities in rivers adapt to changes in the concentration and the composition of organic carbon, reflecting the influence of species sorting [72]. These observations indicate that DOC was strongly associated with point pollution effects due to human activity. In addition, diverse pollutants (e.g., pesticides, insecticides, and heavy metals) from industries, agriculture and untreated household sewage might influence the bacterioplankton community in the middle and lower reaches [16]. The positive effects of bacterioplankton diversity on ecosystem functioning decreased with increasing environmental stress or multiple disturbances.

## 5. Conclusions

The high-throughput sequencing and network analyses of water samples from the Yuan River provided the first evidence that naturally occurring bacterioplankton display network properties characteristic of unstable communities with the intensification of human activities. High human activity intensity in the lower reaches significantly reduced the bacterioplankton diversity, decreased the ecological niche differentiation and destabilized the bacterioplankton community networks during both the wet and dry seasons. Though human activity altered the composition of modules, they preserved their ecological roles in the network, which is of critical importance for *Proteobacteria,*
*Actinobacteria* and *Bacteroidetes* to shape the structure and niche differentiations in the bacterioplankton molecular ecological networks. It was highlighted that the DOC, T, As, ORP and the Chao1 index were the major drivers of these bacterioplankton’s molecular ecological network structure and modules. Although the study was somewhat limited by the lack of replicates in the year, and bacterioplankton network–ecosystem functioning relationships remain unclear in river ecosystems, this work provides important insights into understanding the role of human activity in shaping the structure and function of bacterioplankton communities in inland waters as well as guidance for the realization of more reasonable and effective river management measures in the future. While the relationships between human activity intensity and stable network properties were identified, we encourage that these relationships be further tested in a wider variety of ecosystems, including the soil communities of other terrestrial biomes and microbiomes from aquatic habitats.

## Figures and Tables

**Figure 1 microorganisms-09-01532-f001:**
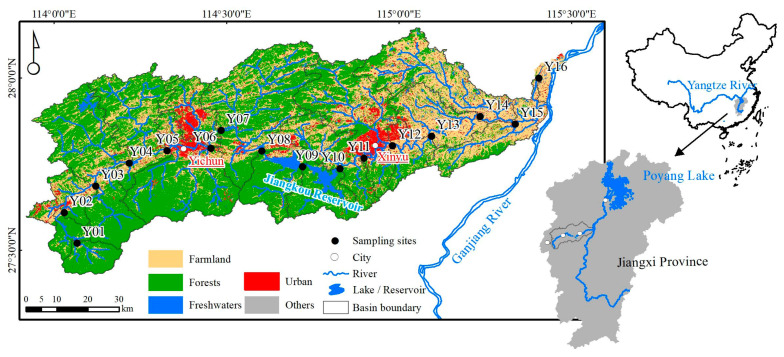
Map of the Yuan River and sampling sites. The cities of Pingxiang, Yichun, and Xinyu are located upstream and downstream in the Yuan River basin.

**Figure 2 microorganisms-09-01532-f002:**
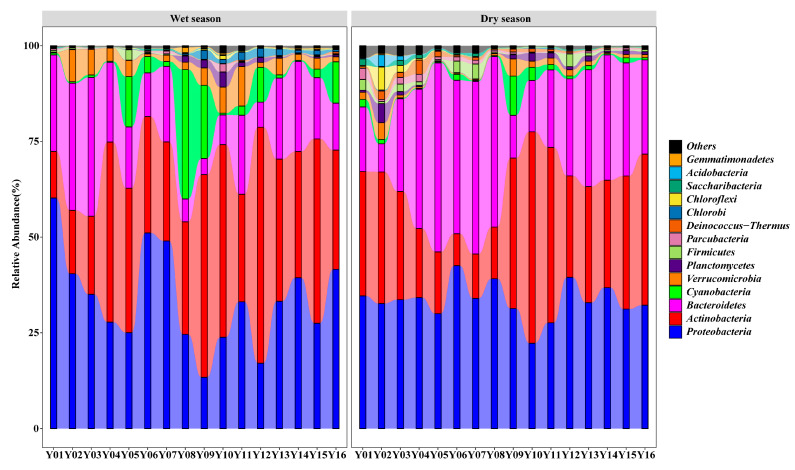
Phylum-level taxonomic composition of the bacterial community under different soil and water conservation measures. Note: the parts with an average abundance of less than 1% were merged and indicated by “others” in the figure.

**Figure 3 microorganisms-09-01532-f003:**
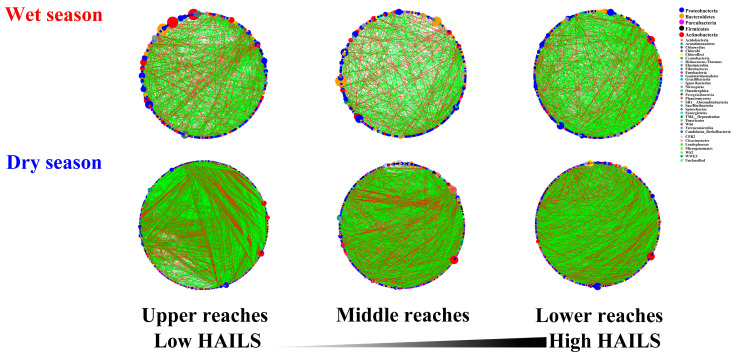
Molecular ecological networks of bacterial communities during the wet and dry seasons on Yuan River. Green and red lines denote significant positive (Spearman correlation, *p* < 0.05, *r* > 0.8) and negative (Spearman correlation, *p* < 0.05, *r* < −0.8) linear relationships, respectively. The size of the circles represents the relative abundances of bacteria and the black line represents the categorical stress rankings of the three environments along the human activity intensity on the land surface.

**Figure 4 microorganisms-09-01532-f004:**
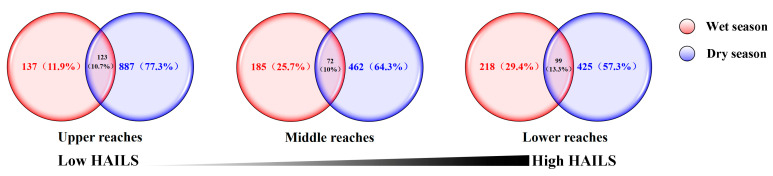
Venn diagram showing the unique and shared nodes among two bacterial molecular ecology networks during the wet and dry season on Yuan River. The black line represents the categorical stress rankings of the three environments along the human activity intensity on the land surface.

**Figure 5 microorganisms-09-01532-f005:**
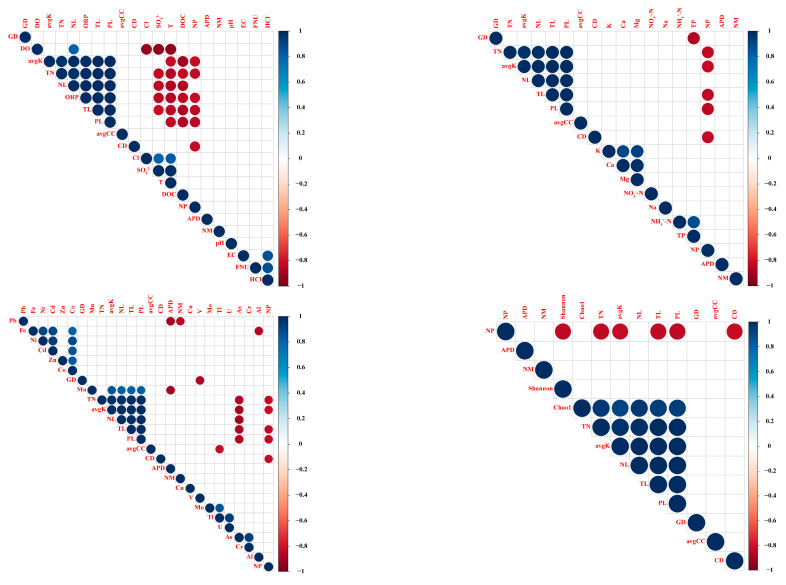
Pearson’s correlation values between network indexes (including GD, avgK, TNs, NLs, TLs, PLs, avgCC, CD, NP, APD and NMs) and the water chemistry parameters (including DO, ORP, Cl^−^, SO_4_^2−^, T, DOC, pH, EC, FUN and HCl^−^), nutrient variables (including K, Ca, Mg, NO_3_^−^-N, Na, NH_4_^+^-N and TP), heavy metals (including Pb, Fe, Ni, Cd, Zn, Co, Mn, Cu, V, Mo, Ti, U, As, Cr and Al) and diversity index (including Shannon index and Chao1 index), respectively.

**Figure 6 microorganisms-09-01532-f006:**
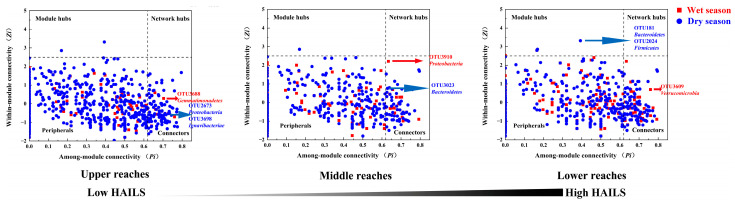
Topological roles of soil fungi during the wet and dry season in the Yuan River.

**Figure 7 microorganisms-09-01532-f007:**
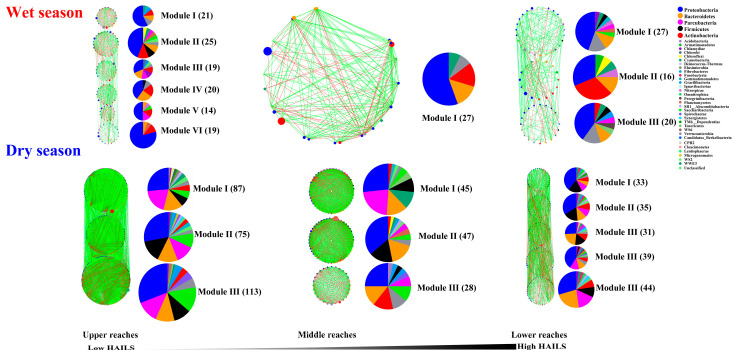
Phylum-level composition of the dominant modules in the bacterioplankton molecular ecological networks in different reaches during the wet and dry season in the Yuan River.

**Table 1 microorganisms-09-01532-t001:** The topological properties of the bacterioplankton molecular ecological networks during the wet season.

Network Indexes	Wet Season		
	Upper Reaches	Middle Reaches	Lower Reaches
Total nodes (TNs)	260	257	317
Total links (TLs)	1389	896	1675
Negative links (NLs)	480	316	493
Positive links (PL)	909	580	1182
Negative/positive (NP)	0.528	0.545	0.417
R square of power-law (R)	0.25	0.639	0.488
Average degree (avgK)	10.685	6.973	10.568
Average clustering coefficient (avgCC)	0.342	0.299	0.264
Average path distance (APD)	3.814	4.509	3.49
Centralization of degree (CD)	0.067	0.071	0.068
Graph density (GD)	0.041	0.027	0.033
Modularity (M)	0.622	0.635	0.505
Number of modules (NMs)	12	31	10

**Table 2 microorganisms-09-01532-t002:** The topological properties of the bacterioplankton molecular ecological networks during the dry season.

Network Indexes	Dry Season		
	Upper Reaches	Middle Reaches	Lower Reaches
Total nodes (TN)	1010	534	524
Total links (TL)	1.964	4638	4811
Negative links (NLs)	4605	1452	1560
Positive links (PLs)	1.4359	3186	3251
Negative/positive (NP)	0.321	0.456	0.480
R square of power-law (R)	0.306	0.431	0.433
Average degree (avgK)	37.552	17.371	18.363
Average clustering coefficient (avgCC)	0.426	0.416	0.221
Average path distance (APD)	3.483	3.857	3.21
Centralization of degree (CD)	0.117	0.075	0.061
Graph density (GD)	0.037	0.033	0.035
Modularity (M)	0.556	0.595	0.410
Number of modules (NMs)	26	29	10

## Data Availability

The data presented in this study are available on request from the corresponding author.

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
