# Peer review of "Spatial and Temporal Distribution of Bacterioplankton Molecular Ecological Networks in the Yuan River under Different Human Activity Intensity"

_microorganisms, 2021, doi:10.3390/microorganisms9071532_

Round 1
Reviewer 1 Report
The article analyzes the influence of human activity and seasonal factors on the bacteriocommunities of a subtropical river. The authors chose a modern metagenomics method to analyze the relationship between the bacteriological community and environmental factors Molecular methods are quite clearly described and applied to isolate bacteriocommunities at various taxonomic levels from species to phylum. On the basis of statistical analysis, the authors also identified the most sensitive taxa of bacterioplankton for changes in anthropogenic impact and traced changes in functional structures. In general, the authors in a broad statistical analysis combined important planes of factors, such as the structure of the landscape, anthropogenic load, chemical parameters of water quality, and molecular analysis of the structure and function of bacterioplankton. The article is written in clear language, although some styling improvements are required in some places. The conclusions of the article are substantiated, the literature data of previous studies and the methods used are provided with a wide bibliographic apparatus. The topic of the article is disclosed and understandable. The article is suitable for publication in the journal Microorganisms, however, some corrections must be made. So, one of the works is in the text, but is not presented in the bibliography. Some care in the arrangement of the cited work numbers should be observed, in particular, the space before the square bracket. The Introduction section can be shortened somewhat. At the end of the introduction, insert a phrase with a hypothesis, and not with a final remark. All these remarks are noted in the attached file of the article. After minor corrections, the article can be accepted for publication.

Reviewer 2 Report
Wu et al. performed an interesting study to investigate the role of the human activity in shaping the structure and function of bacterioplankton molecular ecological networks in subtropical rivers. Further, they provided an insight into the mechanisms of this process, which furnished important information about human-water interaction processes, sustainable uses of freshwater, and watershed management and conservation. Overall, the organization and structure of the manuscript are logical. The manuscript can be accepted for publication. However, I have a few minor comments:
- All the abbreviations should be defined at their first mention in the manuscript.
- All the data should be reported as mean ± S.D.
- Conclusions and abstract should be rewritten to serve their intended purposes. Include major quantitative data also.
